# Bayesian inference of functional asymmetry in the homotrimeric ligand-gated ion channel P2X$_2$
Luciano Moffatt [1] ✉ & Gustavo Pierdominici-Sottile [2]

The trimeric ATP-gated receptor P2X$_2$ offers a minimalist scaffold for dissecting subunit-coupled allosteric activation. Despite closed and open structures, the physical origin of the transient 'flip' intermediate and P2X$_2$'s negative cooperativity remain unresolved. Rapid patch-clamp kinetics parsed by Bayesian inference (MacroIR) and supported by atomistic simulations reveal that P2X$_2$ activates through a sequential, asymmetric coupling mechanism. ATP binding to any inter subunit pocket selectively reduces the rotational barrier on one of the two framing subunits, triggering partial activation of the receptor while the other subunit is minimally affected. This rotation then raises the barrier for the next ATP-binding event, accounting for the receptor's negative cooperativity. Under ligand-free conditions, heightened rotation barriers trap the channel in its closed conformation and quantitatively reproduce the spontaneous current fluctuations we record. These results overturn the canonical view of symmetric, concerted gating in P2X$_2$ and explains the classical flip state as an obligatory structural intermediate.

Ligand-gated ion channels (LGICs) are multimeric membrane proteins that mediate rapid electrical and chemical signalling, underpinning synaptic transmission, muscle contraction, inflammatory processes, cardiovascular responses, and immune defence[1–5]. ATP-activated P2X receptors, with their minimalist trimeric architecture and well-defined inter-subunit ATP pockets, provide an experimentally tractable platform for probing allosteric gating mechanisms[6–9].

Despite detailed structural snapshots of closed and open states, the field has long faced a disconnect between classical kinetic models, often invoking abstract intermediates such as the "flip state"[10,11], and the underlying molecular transitions that generate them. This "flip state" proposed to reconcile rapid ligand binding with slower pore opening, has remained phenomenological, lacking clear structural or energetic grounding. This disconnect is particularly acute in P2X$_2$, where structural symmetry should enforce equivalent subunit behaviour, yet experiments reveal functional asymmetry and negative cooperativity[7]. Because each ATP-binding site bridges two subunits, partial occupancy can bias one rotation over its neighbour, but macroscopic currents blur these effects into coarse, time-averaged signals.

To tackle this problem, we compared nine mechanistic schemes-ranging from classical state-transition and allosteric models to two conformational models that explicitly couple three subunit rotations to ATP occupancy-and identified Scheme IX as the most plausible mechanism. Each scheme was fitted to outside-out macropatch recordings using a recursive Bayesian update (MacroIR, for *Macroscopic Interval Recursive*) that propagates boundary-conditioned priors through every measurement interval, thereby correcting for the unavoidable time averaging of macroscopic currents and yielding high-fidelity interval-averaged likelihood approximations. This recursive approach links fluctuations in macroscopic recordings to mechanistic schemes, remains valid even when integration windows exceed kinetic timescales, and produces Bayesian evidences-the gold standard for comparing candidate models. Posterior distributions of kinetic parameters were then explored using affine-invariant parallel-tempering Markov-chain Monte Carlo, which efficiently samples rugged posterior surfaces and ensures robust convergence. Thermodynamic integration across these tempered likelihoods yielded the final Bayesian evidences.

Complementary microsecond molecular-dynamics simulations of zebrafish P2X$_4$ with a single ATP bound in one pocket showed that the subunit on the "A" side of the ligand consistently initiates rotation before its "B" neighbour, pinpointing the directional preference that electrophysiology alone cannot resolve and corroborating the asymmetric pathway predicted by Scheme IX.

Our findings reveal that functional asymmetry can emerge from the geometry of ligand-induced coupling in the structurally symmetric

[1]Instituto de Química Física de los Materiales, Medio Ambiente y Energía, Consejo Nacional de Investigaciones Científicas y Técnicas, Facultad de Ciencias Exactas y Naturales, Universidad de Buenos Aires, Ciudad de Buenos Aires, Argentina. [2]Departamento de Ciencia y Tecnología, Consejo Nacional de Investigaciones Científicas y Técnicas, Universidad Nacional de Quilmes, Bernal, Buenos Aires, Argentina. ✉e-mail: lmoffatt@qi.fcen.uba.ar

assembly of the trimeric P2X$_2$ receptor; whether similar mechanisms operate in tetrameric or pentameric LGICs remains to be tested.

## Results

### Bayesian selection of kinetic schemes for P2X$_2$ activation

We used Bayesian model evidence to rank nine mechanistic schemes for ATP-induced activation of P2X$_2$ receptors (Fig. 1), spanning classical state models, canonical allosteric models, and two subunit resolved conformational mechanisms. Model evaluation relied on outside-out patch clamp recordings from rat P2X$_2$ channels stimulated with 0.2 ms ATP pulses (0.1–10 mM)[11].

Posterior sampling was performed with an *affine invariant parallel tempering Markov chain Monte Carlo* sampler (PT MCMC)-an ensemble of replicas that evolve at progressively higher "temperatures" and periodically swap states, thereby traversing multimodal posteriors more efficiently than single chain MCMC. PT MCMC was embedded in our recursive MacroIR pipeline, which propagates boundary-conditioned ensemble moments through each integration window to yield high-accuracy interval-averaged likelihoods that respect temporal correlations in macroscopic currents. Model evidences were obtained by thermodynamic integration over the tempered likelihood ladder. As a control, we repeated the analysis with the non-recursive MacroINR likelihood, which ignores time averaging and correlation structure, leading to systematically different evidence values (Supplementary Table S1).

The candidate schemes were grouped as follows:
- **State models:** Discrete closed, flipped, and open states.
- **Allosteric models:** Transition rate modulation without explicit structural mapping.

- **Conformational models:** ATP binding coupled to independent subunit rotations at the binding interface.

A conformational scheme with asymmetric subunit coupling (Scheme IX) was decisively favored, exceeding its symmetric counterpart (Scheme VIII) by a Bayes factor >5000 (Fig. 1d). The top-ranked non-conformational alternative (the subunit-specific allosteric Scheme VI) was disfavored by a factor of 6.4 (90% equal-tail interval: 5.5–6.7). Minimal state models and concerted allosteric schemes performed poorly (Fig. S1).

These findings establish that:
1. Subunit rotation is the principal conformational step in activation;
2. ATP binding allosterically modulates rotational propensity at the interface;
3. This modulation is strongly asymmetric between subunits;
4. Macroscopic conductance scales with the number of rotated subunits.

Notably, the classical "flip state" is reinterpreted here as a probabilistic manifestation of underlying subunit transitions.

Model ranking was sensitive to likelihood approximation: the control method (MacroINR) systematically underestimated evidence for schemes with conformational intermediates, underscoring the importance of rigorous handling of temporal correlations (Fig. 1d–e).

**Computational considerations.** Bayesian model selection at this mechanistic depth is computationally intensive, with each scheme requiring 12–52 days to converge on a 16–32-core cluster. Explicitly convolving the patch-clamp amplifier's low-pass filter kernel (10 kHz

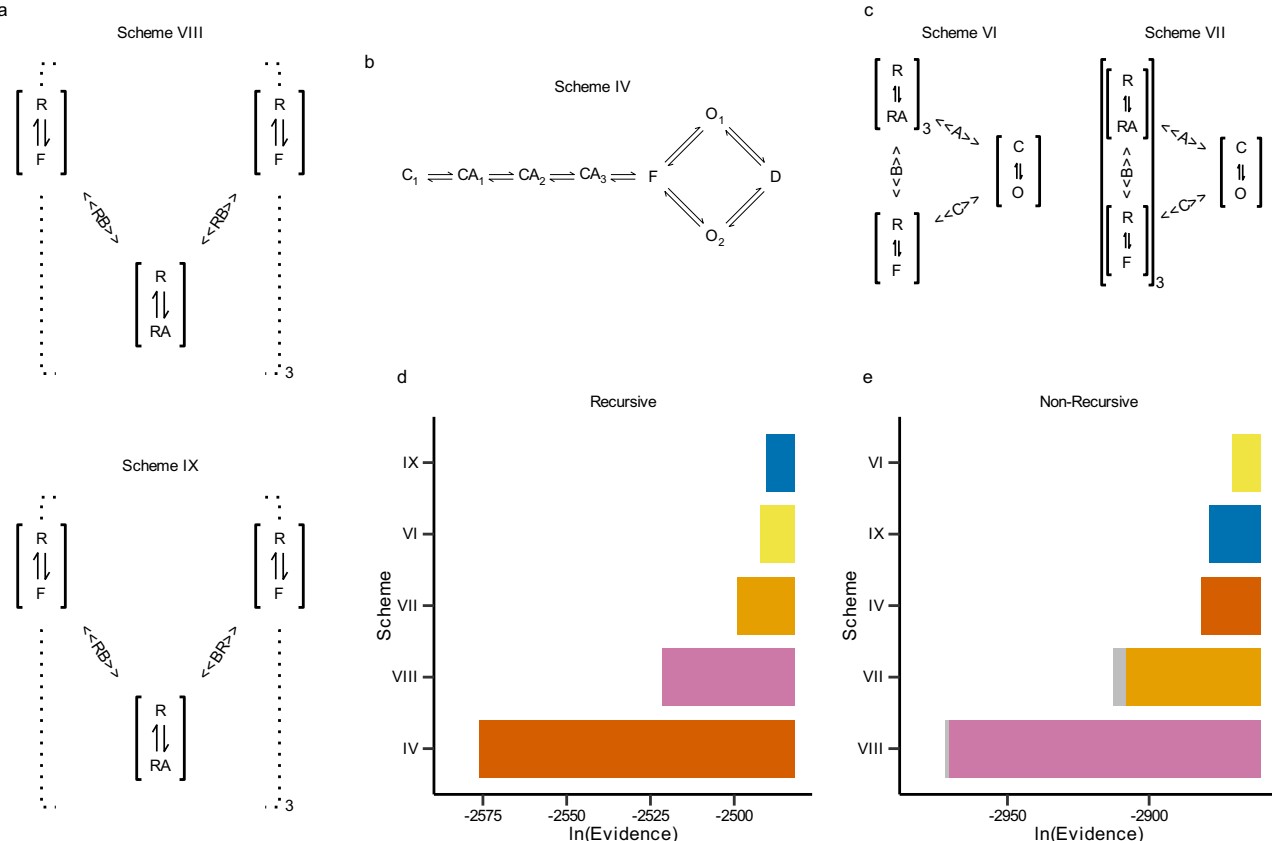

**Fig. 1 | Bayesian evidence ranking via MacroIR and MacroINR. a** Conformational models based on subunit rotation. R resting, F flipped, RA agonist-bound; RB, BR: allosteric coupling via left or right subunits. **b** State model (Scheme IV) with a flip state branching into two open states. **c** Allosteric Schemes VI and VII, featuring concerted vs. subunit-specific flipping. **d** Recursive MacroIR evidence. Log-evidences for each scheme computed by thermodynamic integration over tempered likelihoods in the MacroIR pipeline. **e** Non-recursive MacroINR evidence. Log-evidences computed without temporal correlation corrections, using the MacroINR likelihood approximation. Colors indicate model class; error bars denote standard errors of $\log_e$(Evidence).

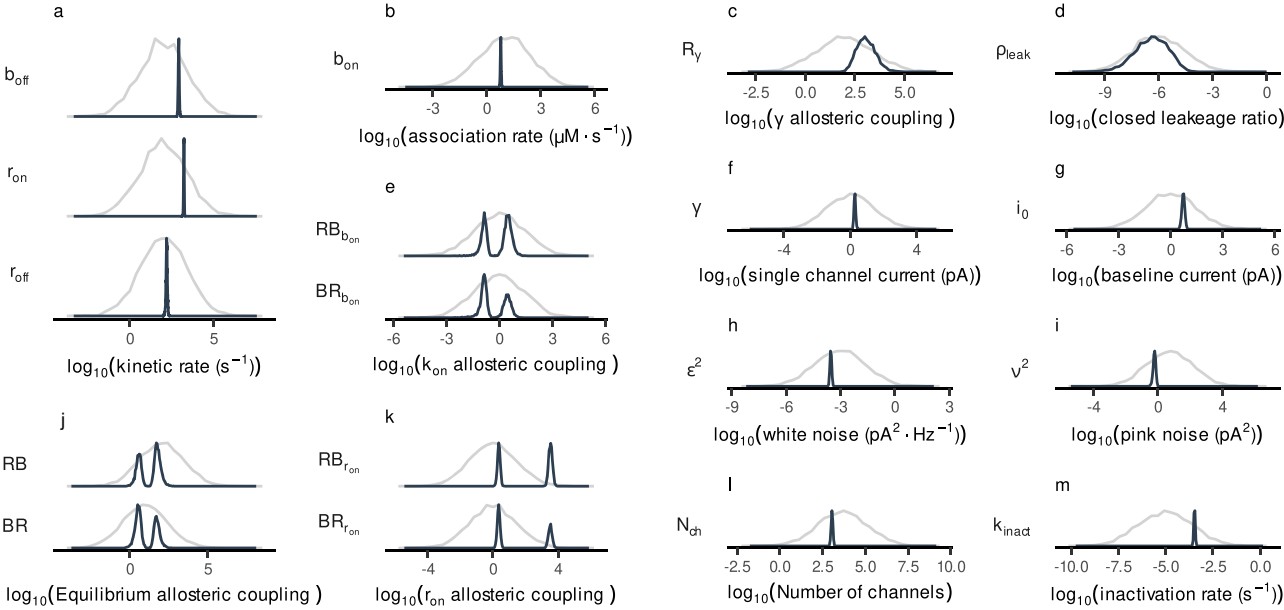

**Fig. 2 | Posterior distributions of model parameters for Scheme IX via MacroIR.** Posterior samples were generated using the interval-update likelihood and PT-MCMC steps of the MacroIR pipeline. **a** Kinetic rate constants: $r_{on}$ (rotation), $r_{off}$ (return to resting state), and $b_{off}$ (unbinding). **b** Association rate: $b_{on}$ (binding). **c** Rotation-conductance coupling factor ($R\gamma$). **d** Closed-channel leakage ratio ($\rho_{leak}$). **e** Allosteric coupling factor affecting the binding rate ($BR_{b_{on}}$, $RB_{b_{on}}$). **f** Single-channel current of the fully rotated channel ($\gamma$). **g** Baseline current ($i_0$). **h** White noise power ($\epsilon^2$). **i** Pink noise power ($\nu^2$). **j** Equilibrium allosteric coupling factors ($BR$, $RB$). **k** Allosteric coupling factors affecting the rotation rate ($BR_{r_{on}}$, $RB_{r_{on}}$). **l** Number of active channels ($N_{ch}$). **m** Inactivation rate ($k_{inact}$). The gray line indicates the prior; the colored line shows the posterior for each parameter. Probability densities are normalized to their maximum for comparability.

Bessel; 50 kHz digitization[12]) would increase computational demands more than tenfold, rendering such analyses impractical at present.

In summary, Bayesian evidence identifies a directional, asymmetrically coupled gating mechanism for P2X₂, motivating further structural and functional investigation of this symmetry breaking.

### Posterior analysis of Scheme IX

The posterior distributions of the parameters in Scheme IX (Fig. 2) fall into three distinct classes, reflecting how different aspects of the data constrain the model:

**Well-identified kinetic and noise parameters.** Core parameters governing ligand binding ($b_{on}$, $b_{off}$), subunit rotation ($r_{on}$, $r_{off}$), channel conductance ($\gamma$), baseline current ($i_0$), inactivation ($k_{inact}$), channel number ($N_{ch}$), and both white and pink noise ($\epsilon^2$, $\nu^2$) all exhibit narrow, unimodal posterior distributions (posterior concentration factors (PCF) ranging from 16 to 68; Supplementary Table S4). This demonstrates that the 0.2 ms ATP-pulse data provide strong constraints on the fundamental kinetic rates and noise sources governing channel activation and deactivation.

**Moderately informed conductance scaling.** Parameters describing how subunit rotation affects current, the closed-channel leakage ratio ($\rho_{leak}$), and the rotation-conductance factor ($R\gamma$) show reproducible shifts from their priors (PCFs of 1.5 and 3, respectively) but retain appreciable uncertainty. The data are sensitive to the principal features of the current-rotation relationship, but do not fully resolve very low levels of resting (leak) current.

**Left-right ambiguity in binding-rotation coupling.** Posterior distributions for the coupling constants that link ATP binding to subunit rotation (and vice versa) are distinctly bimodal (see Fig. 2 panels e, j, k). This bimodality arises because macroscopic currents alone cannot distinguish which subunit, at a given binding pocket, receives the stronger allosteric effect, so the MCMC sampler explores both mirror solutions. Although weakly informative priors can bias the sampler, the kinetic data

preserve this ambiguity, reflecting the underlying symmetry of the homotrimer.

To resolve this degeneracy, we leveraged single-ATP molecular dynamics (MD) simulations (Results). These revealed that, upon ATP binding at an interface, the Left Flipper (LF) and Head domains of chain A and the Dorsal Fin (DF) domain of chain B move furthest toward the open conformation (Fig. 3 and Table 1). Consistent with these structural displacements, we relabeled the posterior samples so that the stronger coupling ($RB$) is always assigned to chain A, and the weaker ($BR$) to chain B. This symmetry-breaking transformation collapses the bimodal posteriors into interpretable, unimodal distributions.

Together, these three posterior patterns validate the kinetic model's internal consistency, highlight which parameters are most strongly constrained by experiment, and set the stage for integrating kinetic and structural analyses to explain the functional basis of gating asymmetry.

### Single-bound Molecular Dynamics simulations clarify coupling asymmetry

To resolve the ambiguity in the binding-rotation coupling parameters identified by kinetic modeling, we ran MD simulations of the closed state receptor with ATP pre positioned in one arbitrarily chosen inter subunit pocket (A-B interface). As all three pockets are energetically equivalent before binding, the interface selection carries no mechanistic bias. These simulations aimed to infer the initial perturbations caused by one ATP molecule. For the structures sampled during the simulations, we quantified its Degree of Closed-to-Open Transition (DCOT) and its alignment with the opening direction (Degree of Closed-to-Open Direction, DCOD).

Figure 3 presents the DCOT analysis revealing the distinct behavioral patterns between the chains A and B triggered by the interaction of ATP in the binding clef they composed. In general, it is observed that the examined regions of chain A move towards the open form in a higher degree than those of the B chain. The comparison between LF(A) and LF(B) (Fig. 3b), for instance, shows that the first one exhibits a distribution markedly shifted toward the open conformation than the latter. This shift of LF(A) generates a substantial alteration in its interaction pattern with DF(B), effectively

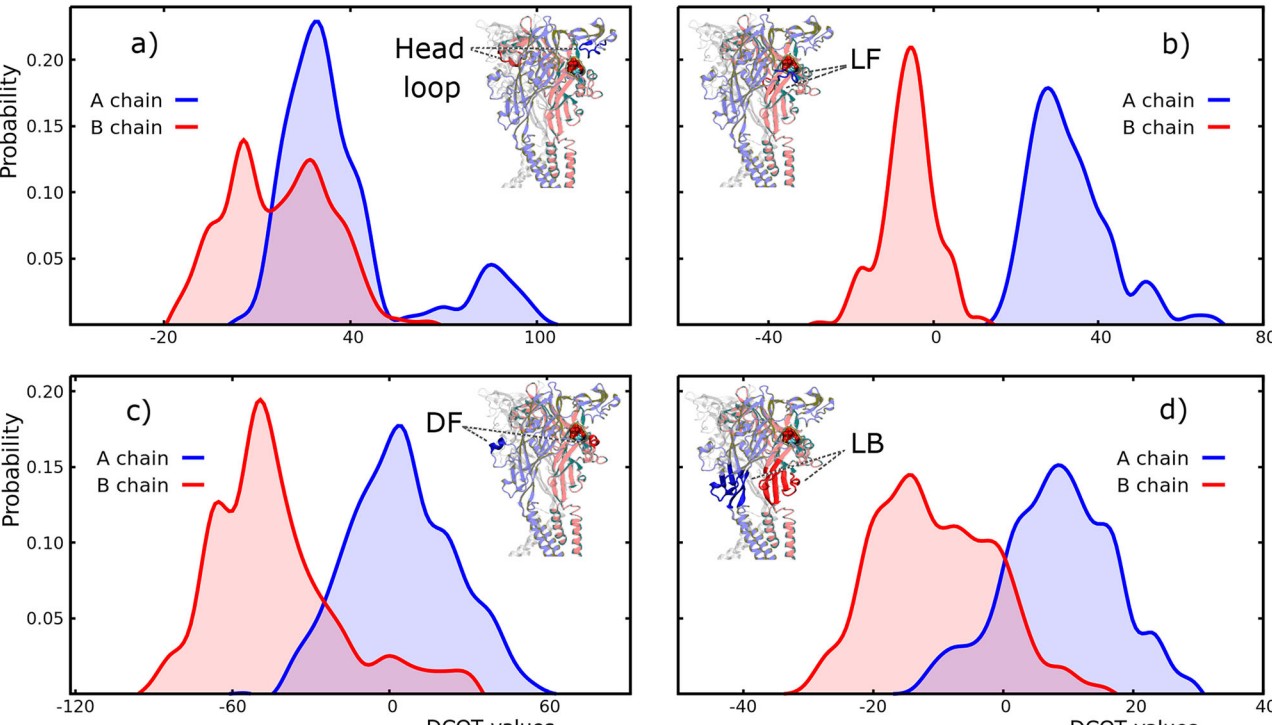

**Fig. 3 | Normalized distribution of DCOT values (See Supplementary Information Eq. (S7)) of distinct regions of $P2X_{1-ATP}^{closed}$.** Chain A is shown in blue, and chain B is shown in red. **a** displays the results for the Head-Loop, **b** for LF, **c** for DF, and **d** for Lower Body (LB). Each panel also includes a schematic illustration of the P2X receptor, with the specific region highlighted for clarity.

**Table 1 | DCOD averages (See Supplementary Information Eq. (S8)) of $P2X_{1-ATP}^{closed}$ for LF, DF, LB and Head-Loop of chains A and B**

| Region | Chain | DCOD average | Standard deviation |
|---|---|---|---|
| Head-Loop | A | 46.04 | 22.12 |
| Head-Loop | B | 5.31 | 5.22 |
| LF | A | 85.83 | 16.16 |
| LF | B | 69.28 | 10.58 |
| DF | A | 72.19 | 19.45 |
| DF | B | 89.02 | 10.07 |
| LB | A | 76.07 | 10.01 |
| LB | B | 86.13 | 4.55 |

triggering an initial "release" of DF(B) from the ATP-bound cleft and promoting a movement in the direction opposite to the typical closed → open transition (Fig. 3c). In addition to the shift in LF distributions favoring chain A toward conformations closer to the open state, the Head-loop and Lower Body (LB) regions further emphasize this asymmetry, with chain A consistently adopting a more open-like conformations (panels a) and d) of Fig. 3). Particularly, the distinct displacements observed in the LB regions of both chains, which are structurally connected to the transmembrane domains[8], suggest an initial asymmetrical channel activation mechanism triggered by the first ATP-binding event.

On the other hand, DCOD analysis (see Table 1) reveals that all examined regions exhibit substantial movement in the closed-to-open direction, regardless of their interaction with ATP. The only exception is the Head-loop of chain B, which shows minimal movement along this pathway direction. This is in line with the fact that the Head domain plays a role in ATP recruitment but is not inherently coupled to the channel opening motion[13,14]. However, when the Head-loop interacts with ATP (as in chain A), its movement shifts markedly toward the open conformation.

A similar analysis was conducted for chain C, with results available in (Fig. S2, DCOT analysis) and in the Supplementary Information Table S4. Overall, the DCOT/DCOD results of chain C regions closely resemble those of the corresponding ones in chains A or B that are not interacting with ATP (Head-loop(C) aligns with Head-loop(B), and DF(C) with DF(A), etc.).

DCOT/DCOD analysis shows that, once ATP is bound, the receptor's first conformational response is markedly asymmetric, fully consistent with the post-binding asymmetric coupling deduced from the kinetic analysis. By comparing MD and kinetic data, we assigned the stronger binding-rotation coupling (RB) to chain A and the weaker coupling (BR) to chain B. This structurally informed relabeling resolved the kinetic model's symmetric degeneracy, collapsing bimodal parameter distributions into interpretable unimodal forms (Fig. S2). In summary, MD simulations validate and clarify the asymmetric allosteric modulation indicated by Bayesian inference, directly linking structural rearrangements to the kinetic asymmetry observed in macroscopic currents. These findings set the stage for a detailed mechanistic interpretation of barrier modulation in the next section.

**Asymmetric modulation of transition barriers by ATP binding**
To understand how ATP binding remodels the energetic landscape underlying $P2X_2$ gating, we modeled three coupled transitions: ligand binding, rotation of the left subunit (subunit A at the A/B binding site), and rotation of the right subunit (subunit B at the same site) (Fig. 4a). Each transition was decomposed into *on* and *off* rate components, explicitly quantifying how ATP occupancy modulates both equilibrium and kinetic barriers. Conventional allosteric models encode coupling with a single equilibrium constant; here, we resolve modulation at the level of individual transition rates, allowing us to distinguish effects on both state stabilization and barrier height.

Posterior distributions (Fig. 4b, c) show that ATP binding strongly accelerates rotation of the left (A) subunit, but has minimal effect on the right (B) subunit. Similarly, rotation of subunit A substantially increases the rate of ATP binding ($RB_{b_{on}} \gg 1$), while rotation of subunit B has a weak or even inhibitory effect ($BR_{b_{on}} < 1$). These patterns reveal a pronounced

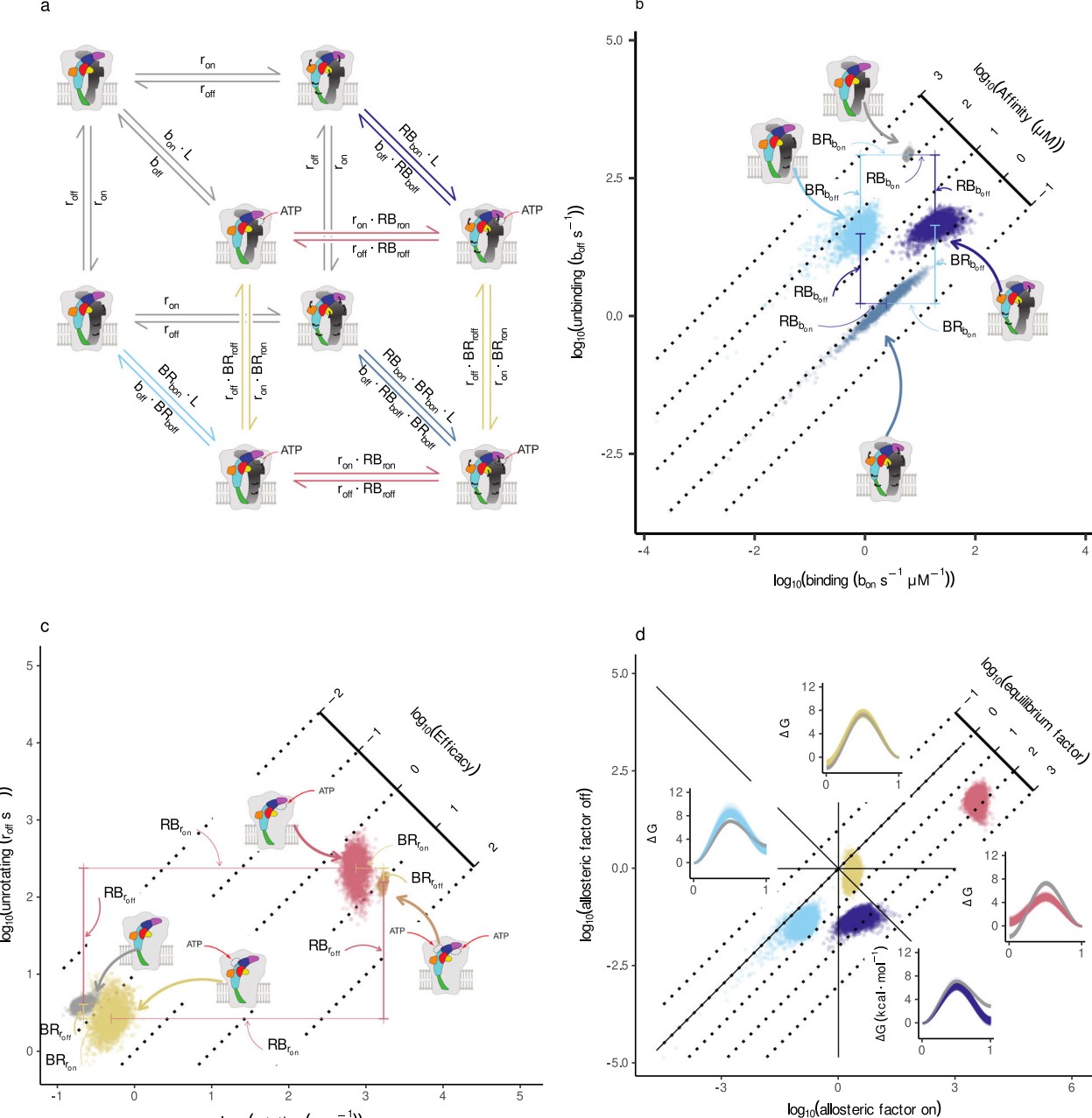

**Fig. 4 | Kinetic barrier modulation via MacroIR inference. a** Cubic schematic illustrating three coupled transitions-agonist (ATP) binding, left subunit rotation, and right subunit rotation-where each vertex represents a distinct receptor state and each edge is labeled with its corresponding kinetic expression. Drawings adapted from ref. 8 underscore the correspondence between model states and channel structures; arrows indicate subunit rotations and ATP occupancy. **b, c** Posterior distributions of binding and rotation rate constants obtained from the interval-update and PT-MCMC steps of MacroIR, segregating into four clusters that reflect the receptor's structural states. **d** Log-log plot of the on versus off rate coupling factors, capturing how ATP binding or subunit rotations differentially accelerate or decelerate kinetic transitions. Insets illustrate changes in energetic barriers, computed with a preexponential factor of $10^{-6}$ s.

functional asymmetry in coupling, in agreement with both the kinetic and MD analyses.

A log-log analysis of coupling factors (Fig. 4d) further distinguishes between mechanistic regimes: (i) *regular allosterism* (stabilizing active states), (ii) *catalytic allosterism* (lowering activation barriers without a major equilibrium shift), and (iii) *inhibitory allosterism* (raising barriers to suppress transitions). ATP's effect on subunit A exemplifies catalytic allosterism, enhancing both *on* and *off* rates, whereas BR coupling is weak, indicating minimal modulation of subunit B.

This asymmetric barrier tuning provides a mechanistic rationale for sequential, ordered subunit activation. By selectively lowering the barrier for rotation of subunit A, ATP binding directs the channel toward productive gating while limiting unproductive or energetically costly transitions. Thus, asymmetric allosteric modulation emerges as a general mechanism for controlling conformational flux and kinetic filtering in the trimeric P2X$_2$ receptor.

In the following section, we quantitatively integrate these energetic insights into a full Markov model, connecting subunit-level transitions to macroscopic gating behavior.

## Integrating subunit conformational dynamics into channel kinetics

To quantitatively link subunit-level conformational dynamics with macroscopic channel behavior, we constructed a Markov model explicitly

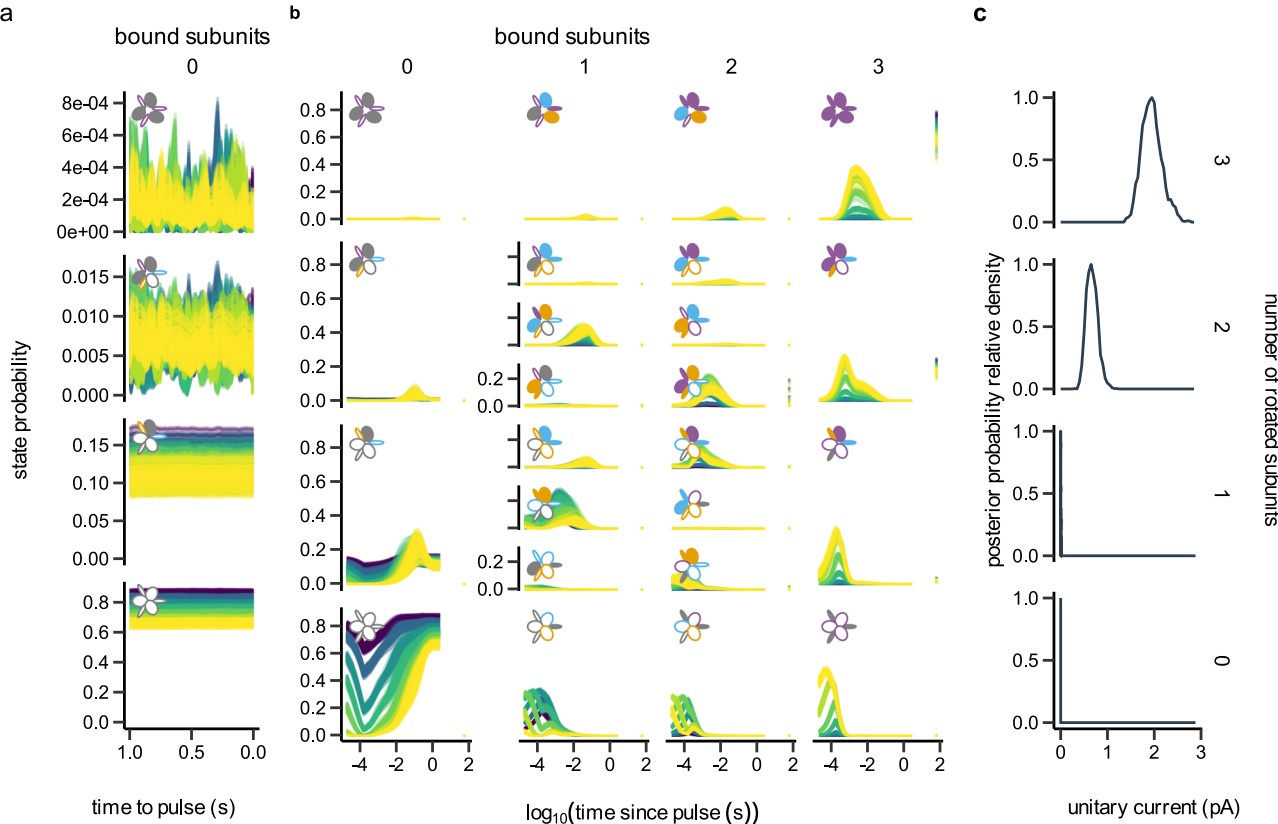

**Fig. 5 | State occupancies and functional currents inferred by MacroIR.**
**a** Equilibrium state occupancies 60 s after stimulus clearance, inferred via the recursive interval-update of MacroIR, revealing significant pre-activation in the absence of ligand. **b** Temporal evolution of state probabilities during 10 mM ATP stimulation, showing sequential activation via ligand binding and subunit rotation as computed by the MacroIR likelihood engine. **c** Distribution of unitary currents across rotational states: channels with two rotated subunits conduct 34% (26–45% equal-tailed interval) of maximal current, explaining baseline currents under ligand-free conditions.

representing all combinations of subunit binding and rotation states (Supplementary Fig. S3). Each subunit undergoes ligand binding/unbinding and rotation/unrotation, with transition rates parameterized by the asymmetric allosteric couplings inferred previously (Fig. 4). Transition rates were scaled by the number of eligible subunits or sites, capturing both combinatorial multiplicity and asymmetric modulation.

This full kinetic framework enables calculation of state occupancies under both resting and stimulated conditions. Under ligand-free conditions (60 s post-ATP removal), the model predicts that stochastic fluctuations drive non-negligible occupancy of partially activated states: ~15% of channels have one subunit rotated, and ~ 1.1% have two (Fig. 5a). Thus, spontaneous baseline currents observed in patch-clamp experiments are quantitatively attributed to transitions into these partially active states, not to measurement noise. This prediction is confirmed by comparison to measured pre-ATP currents (see Fig. 6).

During ATP stimulation (10 mM pulses), state occupancies evolve sequentially: ligand binding precedes subunit rotation, yielding clear edge-to-center transition patterns (Fig. 5b). Deactivation, by contrast, proceeds via mixed ligand-bound and rotated states, highlighting distinct forward and reverse pathways.

Crucially, the model reveals that intermediate states are not merely transient: channels with two rotated subunits conduct 34% of the maximal current (26–45% equal-tailed interval; Fig. 5c), confirming that partial activation is physiologically significant, a result consistent with experimental findings that two occupied binding sites suffice for robust gating[15,16].

These results demonstrate that the observed spontaneous and partial activations are a direct consequence of the underlying Markovian kinetics. By tuning kinetic barriers, especially in the absence of ligand, the channel can minimize time spent in "risky" intermediate states, ensuring stability without sacrificing responsiveness. This provides a rigorous kinetic explanation for macroscopic currents observed in patch-clamp experiments and suggests that selective targeting of specific subunit transitions could provide new strategies for therapeutic modulation in P2X-related diseases.

In the following section, we directly validate the model's predictive accuracy against experimental recordings.

## Validation of the kinetic model by posterior predictive currents

To test the predictive capacity of our kinetic framework, we compared model-generated macroscopic currents with experimental recordings across a range of ATP concentrations. For each ATP pulse, MacroIR computes the expected current by integrating the state occupancies (Fig. 5a, b) with the unitary conductance of each state (Fig. 5c). Samples drawn from the posterior distribution closely reproduce both the amplitude and time course of the measured macroscopic response (Fig. 6a), validating the accuracy of the inferred channel kinetics.

Importantly, the model also quantitatively explains spontaneous current fluctuations observed prior to ATP application, attributing baseline activity to transitions into partially activated states rather than measurement noise alone.

MacroIR further partitions the total observed variance at each time point into three distinct components: (i) the variance arising from stochastic channel gating, modeled by the underlying Markov process; (ii) white noise, reflecting high-frequency, uncorrelated instrumental and thermal noise; and (iii) pink noise, representing slow, low-frequency baseline drift not captured by the channel model (Fig. 6b). During ATP pulses, the variance is dominated by the combination of modeled gating fluctuations and white noise, whereas pink noise becomes more prominent during the pre-pulse period.

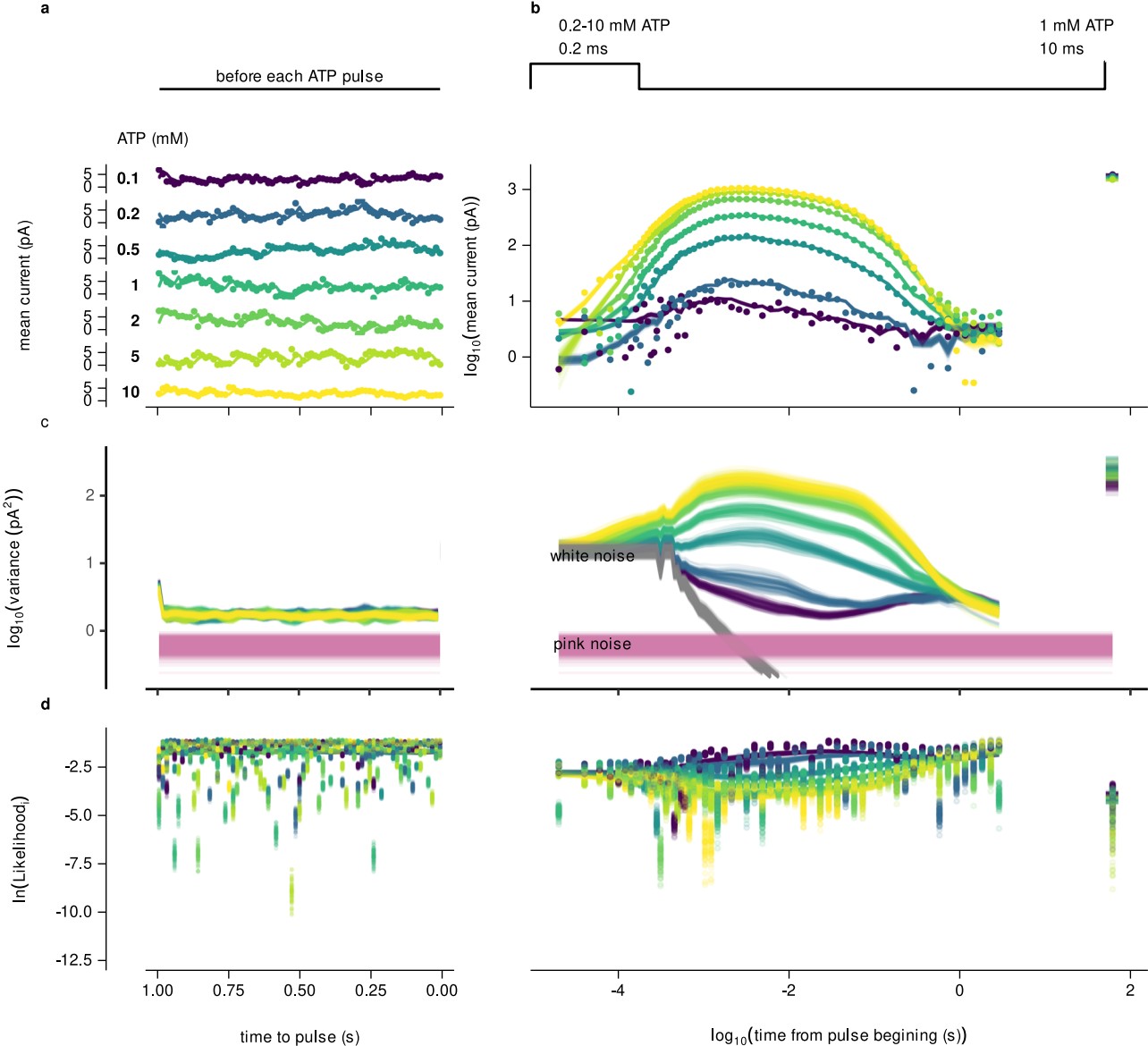

**Fig. 6 | Posterior predictive check using MacroIR-derived state dynamics. a** A sample from the MacroIR posterior predictive distribution of macroscopic current (blue) closely matches the measured current (black). **b** Total variance of the predicted current, with insets showing white (red) and pink (green) noise contributions as estimated by MacroIR; white noise dominates during the ATP pulse, while pink noise is significant during the pre-pulse period. **c** Comparison of expected (lines) and observed likelihoods computed from the MacroIR-predicted normal distribution for each time-averaged measurement.

Finally, direct comparison of the expected and observed likelihoods for each measurement (Fig. 6c) demonstrates excellent agreement between model and experiment, supporting both the validity of the probabilistic framework and the robustness of the parameter inference.

Collectively, these results show that MacroIR not only captures the dynamics and variability of ligand-gated channel activation and deactivation, but also provides a principled, quantitative explanation for spontaneous activity and macroscopic current fluctuations in the absence of ligand. This establishes MacroIR as a robust tool for mechanistic modeling and Bayesian inference in complex signaling systems.

## Discussion

Despite its perfect threefold symmetry, the trimeric P2X receptor does not behave as a functionally equivalent trimer. ATP binding to any of the three symmetry equivalent inter-subunit pockets preferentially lowers the rotational barrier of subunit A, while subunit B is only marginally affected or even slightly disfavoured. The early rotation of subunit A distorts the neighbouring pocket it also helps form, raising the free energy barrier for ATP binding there and thus establishing a structural basis for negative cooperativity[7]. Elevated barriers in the ligand-free receptor stabilize the closed conformation yet still permit rare excursions into partially rotated states, quantitatively accounting for spontaneous openings detected in outside-out patches.

Concatamer studies of $P2X_2$ receptors demonstrate that two ATP-competent binding pockets are sufficient to achieve wild-type unitary currents at saturating ligand concentrations[15,16]. However, concatamer constructs cannot isolate conductance in the absence of ligand. By contrast, our MacroIR Bayesian inference leverages the random fluctuations in ligand-free current (without resolving individual openings) to deconvolve the conductance distribution of defined rotational states. This approach directly estimates the intermediate two-rotation state's conductance at 0.34 (0.26–0.45) $G_{max}$. $P2X_7$ concatamer analyses show that macroscopic ATP-evoked currents persist with only one competent ATP-binding site[17,18]. Taken together, these studies and our analysis indicate that P2X trimers can gate with fewer than three occupied sites: two ($P2X_2$), one ($P2X_7$), or even none (via our MacroIR inference).

ATP modulates *transition barriers* rather than simply shifting equilibria. Such barrier modulation offers a design principle for dynamic proteins: by raising barriers in the unliganded state, the system suppresses wasteful transitions without altering equilibrium distributions, striking a balance between stability and responsiveness in a rugged energy landscape[19,20].

Our favored Scheme IX, containing 24 states dictated by first-principles coupling of binding to independent subunit rotation, is irreducible: removing any transition compromises quantitative fit or violates structural constraints. Bayesian comparison (Bayes factor > 5000 against its symmetric counterpart) decisively selects this asymmetric, sequential mechanism over classical equilibrium models. Posterior analysis further groups parameters into well-identified kinetic and noise terms, moderately informed conductance scalings, and a left-right coupling degeneracy resolved by single-ATP molecular-dynamics simulations that independently revealed larger closed → open displacements in specific domains of chain A.

These mechanistic insights were possible only because MacroIR propagates ensemble moments across arbitrary integration windows, delivering an analytic, high-accuracy interval-averaged likelihood approximation and log-evidence estimates via thermodynamic integration. Earlier approaches either ignored gating noise[21,22], relied on diagonal-covariance approximations[23], idealised each sample as instantaneous[24,25], or reduced each interval to a single distribution point[26].

By completing this methodological arc, MacroIR allowed discrimination among subtle alternatives that lower-resolution analyses could not resolve, even though each model required 12–52 days on a 16–32-core cluster.

Ordered, asymmetric activation narrows the entropic search space, prevents simultaneous risky transitions, and yields graded conductance when not all binding sites are occupied, all features likely to enhance rapid synaptic signalling. Barrier-tuning may therefore govern other multimeric channels and allosteric machines in which partial agonism and negative cooperativity are common[27–30]. Mapping subunit-specific barriers thus opens a path to rational, subunit-selective modulators that tune gating dynamics rather than targeting static conformations[31,32].

Scheme VI fitted almost as well as Scheme IX, and alternative topologies with additional microstates could plausibly capture nuances we did not test. All structural assignments derive from homology models and sub-microsecond simulations of the closed receptor; longer simulations and high-resolution structures of partially liganded states will be needed to confirm the inferred pathway. Although MacroIR rigorously explains spontaneous openings, we cannot exclude residual contributions from other channel populations or experimental artefacts. Single-channel recordings, targeted mutagenesis, and time-resolved cryo-EM will be essential to refine and generalise this kinetic-conformational framework.

By repositioning the flip state as an obligatory consequence of asymmetric barrier modulation, our work reframes allostery in trimeric P2X receptors as dynamic control of transition pathways rather than static equilibrium shifts between states. The integrative strategy presented here combines high-precision electrophysiology, interval-exact Bayesian inference, and structurally informed simulations. Together, these approaches provide a testable hypothesis and methodological template for uncovering latent mechanistic order in trimeric ligand-gated channels, which may guide future experiments across other oligomeric protein assemblies.

## Methods
### Overview
This study integrates electrophysiological recordings, molecular dynamics (MD) simulations, and Bayesian inference to investigate the gating mechanism of the P2X$_2$ receptor. We developed a series of kinetic models, partially informed by MD simulations of the closed state bound to a single ATP molecule, and tested their performance against experimental data using Bayesian model comparison. A central component of our approach is the MacroIR algorithm, which extends our previous work[24] to enable likelihood estimation from time-averaged macroscopic currents. Detailed

mathematical derivations, model specifications (in compliance with Bayesian Analysis Reporting Guidelines), and additional experimental and computational protocols are provided in the Supplementary Information (see SI S1 for full technical details).

### Source and preparation of P2X$_2$ Macro-patch Data
We re-analyzed outside-out patch recordings from our previous study[11], obtained from HEK293 cells expressing rat P2X$_2$ receptors. These high signal-to-noise datasets are ideally suited for kinetic modeling. The stimulation protocol comprised: (i) 0.2 ms ATP pulses at varying concentrations (0.1, 0.2, 0.5, 1, 2, 10 mM) delivered every 2 min; (ii) interleaved 10 ms ATP pulses at 1 mM to monitor response stability and estimate steady-state current; and (iii) precise timing between agonist delivery and current measurement, enabling accurate resolution of rapid activation events. Comprehensive experimental details-including internal and external solutions, temperature, voltage protocol, and equipment-are provided in the original publication[11].

**Active agonist and concentration rationale**. All nominal concentrations refer to *total* MgATP; only the free tetravalent species (ATP$^{4-}$) gates the channel. At pH 7.3 with 1 mM Mg$^{2+}$ and $K_D^{MgATP} \approx 0.2$ mM, approximately 5–8% of total MgATP is free ATP$^{4-}$. For the 0.2 ms pulses, the system is probed in a strict pre-equilibrium regime where the probability that a binding pocket is occupied is

$$P_{bind} \approx 1 - \exp[-k_{on}[ATP^{4-}]\,t].$$

With the posterior median $k_{on} = 6\ \mu M^{-1}s^{-1}$ (Table S3) and $t = 0.0002$ s, half-maximal occupancy requires $[ATP^{4-}] \gtrsim 0.6$ mM, i.e. 5–10 mM total MgATP. Therefore, the 0.1–10 mM MgATP range used[11] spans sub-saturating to near-saturating pre-equilibrium binding.

**Segmentation strategy for kinetic modeling**. To extract kinetic information across multiple timescales, raw current traces were segmented into defined intervals: (i) pre-pulse baseline (1 s), divided into 72 intervals of 13.7 ms and 10 intervals of 0.02 ms to estimate baseline and pink noise; (ii) activation phase, where raw (unaveraged) points were retained during each ATP pulse and for the 0.24 ms immediately preceding onset to resolve fast dynamics; (iii) deactivation phase, where post-pulse decay was downsampled using exponentially increasing intervals, yielding approximately 10 data points per time decade; and (iv) steady-state currents during 10 ms pulses, where a single value was obtained by averaging the final half of each 10 ms pulse response, serving as a proxy for equilibrium current.

This multi-resolution preprocessing yielded the final dataset vectors used in the MacroIR-based Bayesian analysis. Baseline noise was quantified over both 13.7 ms and 0.02 ms windows.

### Kinetic model design and Bayesian comparison
We evaluated nine kinetic schemes, including seven previously established models[11,33] and two newly proposed conformational models inspired by recent structural studies of P2X receptors[8,9]. The new schemes explicitly link ATP binding to partial subunit rotations and introduce either symmetric or asymmetric coupling between binding and gating transitions. In all nine schemes, the three inter subunit ATP pockets share a single association rate constant ($k_{on}$) for the unrotated state; thus, the initial collision probability is fully symmetric, and any asymmetry emerges only after ligand binding.

Models were grouped into three categories: state-based, allosteric, or conformational. A complete description of each scheme-including rate expressions, subunit interactions, and conductance assignments, is provided in Supplementary Information Section S1.2.

Note: Two additional conformational models (previously referred to as Schemes VIII and XI) were excluded from the main analysis after preliminary assessment indicated they did not improve model fit or alter

mechanistic conclusions. Full definitions and analysis of all eleven schemes are provided in the accompanying code and data repository.

## MacroIR inference and thermodynamic-integration evidence

**Workflow overview**.

1. **Prior specification**. Define priors over kinetic parameters and boundary state occupancies, incorporating symmetry constraints and MD guided structural restraints.
2. **Interval update likelihood**. Apply the recursive interval update to approximate the time averaged log likelihood $\ell_t$ for each current segment.
3. **Posterior sampling**. Explore the tempered posterior surfaces with affine invariant parallel tempering MCMC.
4. **Model evidence**. Integrate the tempered log likelihoods via thermodynamic integration to obtain $\log p(\mathbf{D}|\mathcal{M})$ for each candidate scheme.

All experimental measurements of ionic current, no matter how brief the integration time, are inescapably averages over finite intervals, there is no way to record a truly instantaneous current. As a result, time-averaging (whether due to low-pass filtering, hardware integration, or software binning) inevitably induces temporal correlations in the data, violating the assumptions of instantaneous observation and conditional independence that underlie traditional likelihood methods.

To address this, we developed the Macroscopic Interval Recursive (MacroIR) algorithm, a recursive Bayesian framework for inferring kinetic parameters from interval-averaged current data. MacroIR propagates the ensemble mean and covariance of channel state occupancies across arbitrarily long and non-uniform intervals, rigorously conditioning on the state at both the start and end of each interval. By leveraging these "boundary states," MacroIR accurately captures the full statistics of each measurement window, without the need to enumerate all possible underlying state paths.

This boundary-conditioned approach contrasts with previous methods (e.g., Munch et al.[26]), which approximate the likelihood for each interval using only a single-point estimate (typically the mean current and variance at the interval midpoint or average). Such single-point approximations are practical for very short intervals but become unreliable when measurement intervals exceed the timescales of channel kinetics, as in exponentially scaled binning schemes, resulting in biased parameter estimates and underestimation of model evidence.

For comparison, we also implemented a non-recursive alternative, Macroscopic Interval Non-Recursive (MacroINR), which performs single-pass updates without propagating boundary state information. MacroIR thus overcomes the key limitations of prior approaches, enabling accurate and efficient Bayesian inference for kinetic models of arbitrary complexity and across a broad range of timescales.

All mathematical details, code (including validation scripts and usage examples), and a full implementation of MacroIR are provided in our public repository (https://github.com/lmoffatt/macro_dr_submission). Detailed derivations and algorithmic steps are presented in Supplementary Information Section S1.3.

## Molecular dynamics simulations

P2X receptors are homotrimeric or heterotrimeric assemblies, designated $P2X_1$ through $P2X_7$. As of this writing, the Protein Data Bank contains experimental structures from various species for $P2X_1$[34], $P2X_3$[35–39], $P2X_4$[8,9,40,41], and $P2X_7$[42–47]. Despite differences in permeation selectivity among P2X subtypes[48], the overall architecture-especially the extracellular region-is conserved. Each channel resembles a chalice, with each monomer having a dolphin-like shape and binding clefts located approximately 40.0 Å from the membrane surface. This conserved structure strongly suggests a unified closed → open pathway triggered by ATP binding. To date, $P2X_3$, $P2X_4$, and $P2X_7$ are the only subtypes with experimental structures available for both closed and open states[8,9,35,42,44,45].

We investigated the structural basis of asymmetric gating in P2X receptors, considering models from zebrafish $P2X_4$ in its closed and open

states (PDB: 3I5D and 4DW1). These PDBs and this subtype were selected due to the availability of its structures in both states, our extensive experience in conducting robust MD simulations with it[14,49–51], and the conserved architecture shared among all P2X subtypes, which suggests a common dynamic behavior. Particularly, we performed MD simulations on a singly liganded model of the closed state. For this, an ATP molecule was docked into the A-B binding pocket of the closed form of $P2X_4$. Simulations were run in explicit solvent using AMBER, followed by domain-wise analysis of conformational displacements projected onto closed → open transition vectors for four key extracellular regions: Head loop, Left Flipper (LF), Dorsal Fin (DF), and Lower Body (LB).

**Simulation protocols**. MD simulations were performed using the PMEMD module of AMBER24[52]. Each model was solvated in an octahedral box of explicit TIP3P water molecules, with $Na^+$ and $Cl^-$ ions added for charge neutrality and 0.15 M ionic strength. Special care was taken to maintain internal water structure to preserve channel geometry during simulation.

Protein and water were described by the Amber19SB force field[53], while the POPC membrane (where relevant) used Lipid14[54]. The simulation protocol was as follows:

1. Energy minimization (constant volume).
2. Gradual heating (NVT) from 0 K to 100 K over 500 ps.
3. Further heating (NPT) to 303 K. During both heating phases, $C_\alpha$ atoms were restrained with 1.5 kcal/mol Å².
4. Four consecutive equilibration runs of 10 ns each at 303 K, with harmonic restraints reduced stepwise: 0.5, 0.1, 0.05, 0.01 kcal/mol Å².
5. A final unrestrained equilibration of 200 ns.
6. Ten independent production runs of 200 ns each, initiated from the final equilibration snapshot with random Maxwellian velocities at 303 K. Coordinates were saved every 0.05 ns.

Electrostatic interactions were computed with the Particle Mesh Ewald method, applying a 10.0 Å cutoff radius. Thus, direct-space calculations were performed for $r < 10$ Å, while reciprocal-space calculations handled longer-range interactions[55,56]. The SHAKE algorithm was applied to constrain hydrogen-involving bond lengths, enabling an integration time step of 2.0 fs. Stability of results was confirmed by comparing statistics from the first and second halves of each trajectory. Note: Complete simulation input files and data are available in the accompanying GitHub repository (https://gitlab.com/CLPF/p2x).

Details of system preparation and vector analysis are provided in Supplementary Information, Section S1.4.

## Statistics and reproducibility

All statistical analyses were performed within the Bayesian inference framework of MacroIR. Posterior sampling employed affine-invariant parallel-tempering MCMC with 16–32 independent chains, run for 12–52 days until convergence was achieved. Convergence was verified using the potential scale reduction factor ($\hat{R} < 1.01$ for all parameters), effective sample sizes (≥1800), and the Posterior Concentration Factor (PCF).

The PCF is defined as the ratio of prior width to posterior width for each parameter, so that values greater than 1 indicate information gain from the data and higher values reflect stronger parameter identifiability. In some cases, parameters displayed bimodal posterior distributions, which artificially reduced the PCF toward unity. For these, we applied a de-mixing procedure-separating the two posterior modes before recomputing widths-which yielded PCFs of at least 5 in all cases. Parameters governing single-channel conductance remained only modestly constrained (PCF ~1.5–3), whereas the remaining of kinetic and noise parameters were strongly identifiable (PCF ≥15, up to 68). For interpretive guidance: values near 1 denote negligible information gain, 2–5 indicate modest constraint, and values above 10 reflect strong information gain from the data.

Electrophysiological data consisted of outside-out patch recordings from HEK293 cells expressing rat $P2X_2$ receptors, as originally published[11].

Sample sizes ranged from $n = 5$ to $n = 12$ patches per ATP concentration and pulse protocol. Replicates correspond to independent patch recordings from distinct cells. Baseline noise was estimated over $72 \times 13.7$ ms intervals and $10 \times 0.02$ ms intervals per patch. Variance partitioning separated the contributions of stochastic gating fluctuations, white noise, and pink noise, with excellent agreement between modeled and experimental likelihoods.

Molecular dynamics simulations of zebrafish $P2X_4$ receptors were repeated in 10 independent 200 ns trajectories with randomized initial velocities. Structural displacements and closed-to-open transition vectors were reproducible across independent runs, with statistics reported as ensemble means ± standard deviations.

### Reporting summary

Further information on research design is available in the Nature Portfolio Reporting Summary linked to this article.

### Data availability

All data supporting the findings of this study are available within the article and its Supplementary Information, or from public repositories. Processed experimental recordings and MCMC posterior samples are deposited in Zenodo[57] and GitHub (https://github.com/lmoffatt/macro_dr_submission/tree/main/experiments). Molecular dynamics simulation input files for AMBER and analysis scripts are available at https://gitlab.com/CLPF/p2x.

### Code availability

Custom code for kinetic modeling, including all nine mechanistic schemes (and two additional variants not presented here), MacroIR/MacroINR likelihood functions, and MCMC post-processing scripts, is openly available at https://github.com/lmoffatt/macro_dr_submission and has been permanently archived[58]. The core engine is implemented in C++20, while MCMC analysis and figure generation were performed using RMarkdown. There are no restrictions on access.

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

## Acknowledgements

We gratefully acknowledge the computational resources provided by CSC (Buenos Aires), UNC Supercómputo CCAD (Córdoba), and CCAR (Rosario); all part of SNCAD, Argentina. We thank the administrators and technical staff of these facilities for their support. L.M. and G.P-S. are members of CIC-CONICET. This work was supported by Universidad Nacional de Quilmes (PP827-942/20), CONICET, and ANPCyT (ID:2436). The authors also thank Juliana Palma for helpful discussions and feedback.

## Author contributions
L.M. and G.P-S. conceived the study. L.M. performed the kinetic modeling and Bayesian inference. G.P-S. conducted the molecular dynamics simulations and structural analyses. Both authors discussed the results and contributed to the writing of the manuscript.

## Competing interests
The authors declare no competing interests.
