## [Transparent Peer Review file · Communications Biology]

Bayesian inference of functional asymmetry in the homotrimeric ligand-gated ion channel P2X2

Corresponding Author: Dr Luciano Moffatt

This manuscript has been previously submitted at another journal. This document only contains information relating to versions considered at Communications Biology.

Version 0:

Reviewer comments:

Reviewer #1

(Remarks to the Author)

The manuscript from Moffatt and Pierdominici-Sottile describes a novel Bayesian framework applied to patch-clamp recordings augmented by MD simulations to analyze activation of P2X2. They find that activation proceeds via an asymmetric and directional mechanism that explains, for example, a previous observation of negative cooperativity.

I'll be honest, I struggled mightily to follow this paper. I feel that the MacroIR framework is only cursorily introduced with relatively little explanation. This makes it difficult for me to evaluate the application of the method. I feel this lack of clarity continues throughout the whole paper. Thus, I feel without a clearer presentation, I can't adequately judge the work, especially its suitability for this journal. But if other reviewers understand it more readily, I'm willing to concede that the problem is my own.

Reviewer #2

(Remarks to the Author)

The manuscript „Bayesian inference of functional asymmetry in a ligand-gated ion channel“ by Luciano Moffatt and Gustavo Pierdominici-Sottile aimed to analyse the function of a ligand-gated ion channel on the molecular level using a very high quality data set of rat P2X2 receptors and MD simulations combined with a recursive Bayesian framework. It revealed new insights into the sequential binding of ATP followed by partial activation and channel gating. Highlights are the focus on the flipped state, which is difficult to catch by structural studies and the quantitative description of negative cooperativity in ATP binding, which is in general surprising but in the context of channels with binding site between subunits a plausible idea. Support for this idea is coming from a stepwise process where binding of one ATP molecule lowers the rotational barrier for one subunit while raising it for the neighbor at the respective subunit interface. The introduction is compact and gives the reader an entrance to the very well organized results. The methods in combination with the SI are sufficiently described. Existing kinetic schemes for P2X receptors are extended using computationally intensive approaches and very well presented in focusing figures. I enjoyed reading the manuscript and have only minor points.

Although the study is relevant to several ligand-gated ion channels, I would prefer a link to P2X in the title to highlight it. The study shows gating by binding of two ligands to the receptors. Please add the references of studies using concatamers of P2X2 receptors (DOI: 10.1124/mol.112.080903, DOI: 10.1085/jgp.201411166) Furthermore, a discussion of results for P2X7 receptors (OI: 10.1016/j.bbrc.2021.06.101, DOI: 10.3390/cells14120855) could add some value.

The study shows reduced unitary conductance for channels occupied by two ligands. This is in contrast to results from DOI: 10.1124/mol.112.080903. Please discuss the discrepancy.

The main text references work for P2X4 receptors, whereas the SI includes the new structures. Please add the new references in the main text.

Overall this study is a nice example for combining MD simulation and Markovian modelling for P2X receptors and increases insights of their work on the molecular level.

Reviewer #3

(Remarks to the Author)

Version 1:

Reviewer comments:

Reviewer #1

(Remarks to the Author)

I sincerely appreciate the authors' care in addressing my issue with the clarity of the presentation. I am happy to support publication now.

Reviewer #2

(Remarks to the Author)

The authors did a great job. The ms is substantially improved. I do not have further concerns.

Reviewer #3

(Remarks to the Author)

After the revision, the manuscript has been improved and it may be accepted for publication in its current version.

Response to Reviewer 1 Comments

Title: Bayesian inference of functional asymmetry in the homotrimeric ligand-gated ion channel P2X2

Reviewer 1 General commentary

Comment: “The manuscript from Moffatt and Pierdominici-Sottile describes a novel Bayesian framework applied to patch-clamp recordings augmented by MD simulations to analyze activation of P2X2. They find that activation proceeds via an asymmetric and directional mechanism that explains, for example, a previous observation of negative cooperativity.”

Response: We thank the reviewer for this accurate summary.

Reviewer 1 Comment

Comment

“I’ll be honest, I struggled mightily to follow this paper. I feel that the MacroIR framework is only cursorily introduced with relatively little explanation. This makes it difficult for me to evaluate the application of the method. The lack of clarity continues throughout the whole paper; I therefore cannot adequately judge the work. Without a clearer presentation, I can’t adequately judge the work, especially its suitability for this journal.”

Response

We thank the reviewer for this candid assessment. Your main concern was that the *MacroIR* framework was not introduced clearly enough, which in turn made it difficult to follow later analyses.

The idea behind the development of *MacroIR* is to establish a quantitative link between the fluctuations present in macroscopic recordings and mechanistically based kinetic schemes. *MacroIR* stands for *Macroscopic Interval Recursive*, because it allows us to approximate the likelihood function of interval-averaged currents. This is achieved by considering the probability density of the states at both the beginning and the end of each measurement interval. This approach makes it possible to analyze intervals longer than the characteristic time constant of ion channel ensemble kinetics (which scales inversely with the number of channels in the ensemble), thereby opening the possibility of testing more complex kinetic schemes. By employing powerful Bayesian techniques—specifically affine-invariant, parallel-tempering Markov-chain Monte Carlo—we can efficiently explore the parameter space, calculate posterior distributions of kinetic parameters, and evaluate

the evidence of each model (defined as the integral of the likelihood over the parameter space). The evidence is widely regarded as the gold standard for model comparison.

To make *MacroIR* more accessible to readers, we have implemented a series of focused edits that (i) introduce the pipeline in plain language in the *Introduction*, (ii) provide an explicit four-step roadmap in the *Methods*, (iii) expanded the explanation on PT-MCMC in methods and (iv) add concise cues in figure captions and the *Results*. These changes are detailed below. We agree that a clearer up-front description was needed.

We have therefore:

1. **Expanded the *Introduction*.**

Original wording:

Here, we bridge this conceptual gap by integrating legacy outside-out patch-clamp recordings [11] with a recursive Bayesian inference framework (*MacroIR*), enabling rigorous kinetic modeling from time-averaged macroscopic currents.

Revised wording:

To tackle this problem, we compared nine mechanistic schemes—ranging from classical state-transition and allosteric models to two conformational models that explicitly couple three subunit rotations to ATP occupancy—and identified Scheme IX as the most plausible mechanism. Each scheme was fitted to outside-out macropatch recordings using a recursive Bayesian update (*MacroIR* algorithm) that propagates boundary-conditioned priors through every measurement interval, thereby correcting for the unavoidable time averaging of macroscopic currents and yielding high-fidelity interval-averaged likelihood approximations. This recursive approach links fluctuations in macroscopic recordings to mechanistic schemes, remains valid even when integration windows exceed kinetic timescales, and produces Bayesian evidences—the gold standard for comparing candidate models. Posterior distributions of kinetic parameters were then explored using affine-invariant parallel-tempering Markov-chain Monte Carlo, which efficiently samples rugged posterior surfaces and ensures robust convergence. Thermodynamic integration across these tempered likelihoods yielded the final Bayesian evidences.

This paragraph (lines 102–118) now:

- states the practical problem (time-averaged macroscopic currents),
- lists the nine candidate schemes and highlights Scheme IX,

- explains—in one sentence each—(i) the recursive likelihood update, (ii) PT-MCMC posterior sampling, and (iii) thermodynamic-integration evidence,
- explicitly clarifies how MacroIR links fluctuations to mechanistic schemes, why it remains accurate for long integration windows, and why Bayesian evidence is the standard for model comparison.

2. **Renamed and front-loaded the relevant *Methods* subsection.**

Original subtitle:

Recursive Likelihood Estimation with MacroIR

All experimental

Revised subtitle and explanation:

MacroIR inference and thermodynamic–integration evidence

Workflow overview.

- Prior specification.** Define priors over kinetic parameters and boundary-state occupancies, incorporating symmetry constraints and MD-guided structural restraints.
- Interval-update likelihood.** Apply the recursive interval update to approximate the time-averaged log-likelihood ℓ_t for each current segment.
- Posterior sampling.** Explore the tempered posterior surfaces with affine-invariant parallel-tempering MCMC.
- Model evidence.** Integrate the tempered log-likelihoods via thermodynamic integration to obtain $\log p(\mathbf{D} \mid \mathcal{M})$ for each candidate scheme.

All experimental...

The header now reads “*MacroIR inference and thermodynamic-integration evidence*”. Immediately under the header we insert a four-bullet “Workflow overview” covering: prior specification, interval-update likelihood, PT-MCMC, and evidence calculation (no new mathematics added). This one-glance roadmap addresses the reviewer’s request to “properly introduce MacroIR.”

3. Defined **PT-MCMC** at first use in *Methods*, explaining that replicas at different temperatures exchange states to traverse multimodal posteriors.

Original text:

Inference was carried out via affine-invariant parallel tempering MCMC within our recursive MacroIR framework, which rigorously accounts for temporal correlations in macroscopic current data.

Revised and expanded text:

Posterior sampling was performed with an *affine-invariant parallel-tempering Markov-chain Monte Carlo* sampler (PT-MCMC)—an ensemble of replicas that evolve at progressively higher “temperatures” and periodically swap states, thereby traversing multimodal posteriors more efficiently than single-chain MCMC. PT-MCMC was embedded in our recursive MACROIR pipeline, which propagates boundary-conditioned ensemble moments through each integration window to yield high-accuracy interval-averaged likelihoods that respect temporal correlations in macroscopic currents. Model evidences were obtained by thermodynamic integration over the tempered likelihood ladder.

4. Added concise cues in captions and text.

The first result paragraph now says “Using the interval-update of MacroIR (Eq. 3)...”. Figure 1 caption titles panel (d) “Recursive MacroIR evidence,” and Figure 2 caption begins “Posterior distributions . . . via MacroIR.” This ensures that every major result is explicitly linked to the relevant step of the framework.

We believe the above revisions now provide the clarity required to evaluate both the methodological advance and the biological insight:

1. The Introduction lays out the complete pipeline in non-specialist language.
2. Methods supply an easy-to-navigate roadmap plus full mathematical detail.
3. Results and Figures explicitly reference the MacroIR steps from which they derive.
4. The Discussion succinctly states the broader applicability of the framework.

We hope these targeted changes resolve your concerns and make the manuscript fully accessible to the journal’s readership. We are grateful for your feedback, which has substantially improved the clarity of the paper.

Response to Reviewer 2 Comments

Manuscript ID: COMMSBIO-25-6434-T

Title: Bayesian inference of functional asymmetry in the homotrimeric ligand-gated ion channel P2X2

Dear Reviewer, thank you for your constructive feedback. Below we address each of your comments in turn.

Reviewer 2 General commentary

Comment: “The manuscript “Bayesian inference of functional asymmetry in a ligand-gated ion channel” by Luciano Moffatt and Gustavo Pierdominici-Sottile aimed to analyse the function of a ligand-gated ion channel on the molecular level using a very high quality data set of rat P2X2 receptors and MD simulations combined with a recursive Bayesian framework. It revealed new insights into the sequential binding of ATP followed by partial activation and channel gating. Highlights are the focus on the flipped state, which is difficult to catch by structural studies and the quantitative description of negative cooperativity in ATP binding, which is in general surprising but in the context of channels with binding site between subunits a plausible idea. Support for this idea is coming from a stepwise process where binding of one ATP molecule lowers the rotational barrier for one subunit while raising it for the neighbor at the respective subunit interface. The introduction is compact and gives the reader an entrance to the very well organized results. The methods in combination with the SI are sufficiently described. Existing kinetic schemes for P2X receptors are extended using computationally intensive approaches and very well presented in focusing figures. I enjoyed reading the manuscript and have only minor points.”

Response: Dear Reviewer, thank you for your constructive feedback. Below we address each of your comments in turn.

Reviewer 2 Comment 1

Comment: “Although the study is relevant to several ligand-gated ion channels, I would prefer a link to P2X in the title to highlight it.”

Response: We have revised the title to:

Bayesian inference of functional asymmetry in the homotrimeric ligand-gated

ion channel P2X2.

This change directly incorporates the requested P2X identifier into the title while maintaining the concise focus of the study.

Reviewer 2 Comments 2 and 3

Comment:

“The study shows gating by binding of two ligands to the receptors. Please add the references of studies using concatamers of P2X2 receptors (DOI: 10.1124/mol.112.080903, DOI: 10.1085/jgp.201411166). Furthermore, a discussion of results for P2X7 receptors (DOI: 10.1016/j.bbrc.2021.06.101, DOI: 10.3390/cells14120855) could add some value.”

Comment:

“The study shows reduced unitary conductance for channels occupied by two ligands. This is in contrast to results from DOI: 10.1124/mol.112.080903. Please discuss the discrepancy.”

Response:

References to the requested P2X2 concatamer studies have been added and are now discussed in a new paragraph of the *Discussion* (p. 14, lines 28–37; see below) that incorporates the latest P2X7 concatamer data alongside our P2X2 findings and address the discrepancy in unitary conductance. In brief, concatamer constructs with two wild-type binding sites can still propagate rotation to all three subunits at saturating ATP, yielding full-conductance openings, whereas our MacroIR inference uniquely leverages ligand-free current fluctuations to deconvolve and estimate the conductance of the exact two-rotation intermediate ($0.34 G_{\max}$). The full text of the inserted paragraph is provided below.

New paragraph:

Concatamer studies of P2X2 receptors demonstrate that two ATP-competent binding pockets are sufficient to achieve wild-type unitary currents at saturating ligand concentrations [36,37]. However, concatamer constructs cannot isolate conductance in the absence of ligand. By contrast, our MacroIR Bayesian inference leverages the random fluctuations in ligand-free current (without resolving individual openings) to deconvolve the conductance distribution of defined rotational states. This approach directly estimates the intermediate two-rotation state’s conductance at 0.34 (0.26 – 0.45) G_{\max} . P2X7 concatamer analyses show that macroscopic ATP-evoked currents persist with only one competent ATP-binding site [38,39]. Taken together, these studies and our analysis indicate that P2X trimers can gate with fewer than three occupied sites: two (P2X2), one (P2X7), or even none (via our MacroIR inference).

Reviewer 2 Comment 4

Comment: “The main text references work for P2X₄ receptors, whereas the SI includes the new structures. Please add the new references in the main text.”

Response: In the revised version, we have incorporated the description of the new structures into the main text, along with the corresponding references. The sentences introduced in the manuscript are described below.

New paragraph:

P2X receptors are homotrimeric or heterotrimeric assemblies, designated P2X₁ through P2X₇. As of this writing, the Protein Data Bank contains experimental structures from various species for P2X₁ [15], P2X₃ [16–20], P2X₄ [8, 9, 21, 22], and P2X₇ [23–28]. Despite differences in permeation selectivity among P2X subtypes [29], the overall architecture—especially the extracellular region—is conserved. Each channel resembles a chalice, with each monomer having a dolphin-like shape and binding clefts located approximately 40.0 Å from the membrane surface. This conserved structure strongly suggests a unified closed→open pathway triggered by ATP binding. To date, P2X₃, P2X₄, and P2X₇ are the only subtypes with experimental structures available for both closed and open states[8, 9, 16, 23, 25, 26].

Reviewer 2 Comment 5

Comment: “Overall this study is a nice example for combining MD simulation and Markovian modelling for P2X receptors and increases insights of there work on the molecular level.”

Response: We thank the reviewer for this kind observation.

Response to Reviewer 3 Comments

Manuscript ID: COMMSBIO-25-6434-T

Title: Bayesian inference of functional asymmetry in the homotrimeric ligand-gated ion channel P2X2

Reviewer 3 General commentary

Comment: “Structurally, ligand-gated ion channels (LGICs) typically assemble as trimeric, tetrameric, or pentameric complexes. This study focuses on the ATP-gated P2X receptor, a prototypical trimeric channel, to explore the mechanisms of allosteric activation. While static, high-resolution structures have delineated the closed and open conformations, the cooperative interactions among the three ATP-binding events remain insufficiently understood. To address this gap, the authors employ a MacroIR-based approach in combination with molecular dynamics simulations and propose a model featuring directional and asymmetric coupling. While the findings offer valuable mechanistic insights into P2X receptor gating, the study is constrained by several notable methodological limitations.”

Response: We would like to thank the reviewer for this constructive feedback. Below we address each of your comments in turn.

Reviewer 3 Comment 1

Comment: “1. The authors utilize molecular dynamics simulations based on P2X receptor structures; however, they chose the structurally unresolved P2X2 subtype for electrophysiological experiments, despite the availability of high-resolution structures for P2X1, P2X3, P2X4, and P2X7. Similarly, for ATP-binding simulations, they employ the zebrafish P2X4 (zfP2X4) structure, rather than P2X3 or P2X7, which offer both open and closed conformations. These choices lack sufficient scientific justification.”

Response:

Thank you for this important comment. In the revised manuscript, we have added a detailed explanation regarding our choice of receptor structures for both the electrophysiological experiments and molecular dynamics simulations. Specifically:

The electrophysiological data were originally collected in an earlier work, on P2X2, the most studied purinergic receptor due to its accessible kinetics, i.e. slow desensitization.

Regarding the choice of zfP2X4 for the molecular dynamics (MD) simulations, we have now explicitly clarified the rationale in the main text. In summary, the currently available crystal structures of P2X channels include P2X₁, P2X₃, P2X₄, and P2X₇. Among these, P2X₃, P2X₄, and P2X₇ have structures representing both closed and open conformational states. Notably, P2X₄ was the first subtype for which structures of both states were resolved, making it a well-characterized model for studying P2X channel gating mechanisms [8,9].

Our prior experience with zfP2X4 has enabled us to establish robust and reliable MD simulation protocols for this system [30-33], which was a key factor in selecting it as the model to investigate ATP-induced perturbations that drive the closed-to-open transition. Furthermore, all P2X subtypes share a conserved overall architecture—particularly within the extracellular domain—implying a common dynamic behavior and a shared transition pathway between closed and open states. Consequently, insights gained from zfP2X4 simulations are expected to be transferable to the entire P2X receptor family. The following outlines the description added to the manuscript.

New paragraph:

P2X receptors are homotrimeric or heterotrimeric assemblies, designated P2X₁ through P2X₇. As of this writing, the Protein Data Bank contains experimental structures from various species for P2X₁ [15], P2X₃ [16–20], P2X₄ [8, 9, 21, 22], and P2X₇ [23–28]. Despite differences in permeation selectivity among P2X subtypes [29], the overall architecture—especially the extracellular region—is conserved. Each channel resembles a chalice, with each monomer having a dolphin-like shape and binding clefts located approximately 40.0 Å from the membrane surface. This conserved structure strongly suggests a unified closed→open pathway triggered by ATP binding. To date, P2X₃, P2X₄, and P2X₇ are the only subtypes with experimental structures available for both closed and open states [8, 9, 16, 23, 25, 26].

We investigated the structural basis of asymmetric gating in P2X receptors considering models from zebrafish P2X₄ in its closed and open states (PDB: 3I5D and 4DW1). These PDBs and this subtype were selected due to the availability of its structures in both states, our extensive experience in conducting robust MD simulations with it [30-33], and the conserved architecture shared among all P2X subtypes, which suggests a common dynamic behavior.

Reviewer 3 Comment 2

Comment: “2. LGICs span a range of oligomeric forms—trimeric (e.g., P2X, ASIC), tetrameric (e.g., TRP, Kv, NMDA), and pentameric (e.g., nAChR, 5-HTR)—with varying degrees of structural symmetry. A twist-to-open gating mechanism involving cooperative

transitions has been proposed for several subtypes. The current study introduces an asymmetric, negatively cooperative activation model for P2X2, challenging the canonical paradigm of symmetric, cooperative gating. However, extrapolating these findings to other LGICs is premature and potentially misleading, as trimeric receptors cannot be assumed to reflect the mechanistic diversity of all LGIC families.”

Response: We fully agree with the reviewer’s caution regarding premature generalization. We carefully revised the manuscript to limit explicitly the scope of our findings to the trimeric P2X2 receptor. Specifically, we modified the following four passages:

- Abstract (last two sentences):

Original wording:

These findings overturn the prevailing assumption of symmetric, concerted activation, and demonstrate that the classical flip state arises as a necessary physical intermediate. By showing that ligand-induced modulation of activation barriers can drive symmetry breaking in homomeric channels, our results establish a general principle for dynamic protein assemblies, and provide a conceptual basis for designing conformation-selective modulators targeting pain and inflammation.

Revised wording:

These results overturn the canonical view of symmetric, concerted gating in P2X2 and establish the classical flip state as an obligatory structural intermediate.

- Introduction (last sentence):

Original wording:

Our findings reveal that functional asymmetry can emerge from the geometry of ligand-induced coupling in a structurally symmetric assembly, establishing a general principle for dynamic control in multimeric protein machines.

Revised wording:

Our findings reveal that functional asymmetry can emerge from the geometry of ligand-induced coupling in the structurally symmetric assembly of the trimeric P2X2 receptor; whether similar mechanisms operate in tetrameric or pentameric LGICs remains to be tested.

- Discussion (final paragraph):

Original wording:

By repositioning the flip state as an obligatory consequence of asymmetric barrier modulation, our work reframes allostery in dynamic proteins as control of transition pathways rather than static state populations. The integrative strategy presented here combines high-precision electrophysiology, interval-exact Bayesian inference, and structurally informed simulations. Together, these approaches provide a blueprint for uncovering latent mechanistic order in complex protein assemblies and for guiding mechanism-based therapeutics.

Revised wording:

By repositioning the flip state as an obligatory consequence of asymmetric barrier modulation, our work reframes allostery in trimeric P2X receptors as dynamic control of transition pathways rather than static equilibrium shifts between states. The integrative strategy presented here combines high-precision electrophysiology, interval-exact Bayesian inference, and structurally informed simulations. Together, these approaches provide a testable hypothesis and methodological template for uncovering latent mechanistic order in trimeric ligand-gated channels, which may guide future experiments across other oligomeric protein assemblies.

Additionally, we identified one further passage in the Results section that required similar attention. The original statement read:

Thus, asymmetric allosteric modulation emerges as a general mechanism for controlling conformational flux and kinetic filtering in dynamic proteins.

We revised this statement explicitly to limit the generalization:

Thus, asymmetric allosteric modulation emerges as a mechanism for controlling conformational flux and kinetic filtering in the trimeric P2X2 receptor.

We believe these revisions fully address Reviewer 3's valid concerns and clearly position our findings within their appropriate context.

Reviewer 3 Comment 3

Comment: “3. Multiple ATP-binding scenarios are plausible for P2X2. The authors propose a model in which one site exhibits a higher probability of ATP collision, initiating allosteric modulation of the remaining sites. However, they fail to consider an alternative, equally plausible model involving equivalent binding probabilities and cooperative dual-site engagement. Their reliance on a single ATP-binding simulation using zfp2X4 (apo-ATP) lacks sufficient resolution to exclude such alternatives.”

Response: We thank the reviewer for this insightful point. In our framework, **all three inter-subunit ATP pockets share the same association-rate constant** (k_{on}), so the *collision* probability is strictly symmetric; any functional asymmetry arises *only after* ATP binds, via differential coupling of the bound ligand to subunit rotations.

The “equivalent binding probabilities + cooperative dual-site engagement” scenario corresponds exactly to **Scheme VIII** (symmetric coupling) in our model set, which we explicitly fitted alongside its asymmetric counterpart (Scheme IX) and seven other variants. Bayesian model comparison **decisively** favors Scheme IX over Scheme VIII by $\Delta \ln \text{Evidence} \approx 8.5$ (Bayes factor $> 5 \times 10^3$), well above the “decisive” threshold. No additional MCMC runs are therefore required.

We have also identified several passages where our wording could unintentionally imply unequal collision rates. Below, we quote the original manuscript text and our proposed revisions.

Abstract (lines 8–10) Original wording:

ATP binding selectively lowers the energetic barrier for rotation of one subunit at the binding interface, promoting partial activation and substantial conductance, while minimally affecting its neighbor.

Revised wording:

ATP binding to any inter-subunit pocket selectively reduces the rotational barrier on one of the two framing subunits, triggering partial activation of the receptor while the other subunit is minimally affected.

Discussion (first paragraph, second sentence) Original wording:

We show that ATP binding at a single inter-subunit pocket selectively lowers the rotational barrier of one subunit while minimally affecting, and in fact slightly

disfavoring, its neighbour. Once this first subunit rotates, the barrier for a second ATP to bind at the same interface rises, providing a direct mechanism for negative cooperativity and for the phenomenological “flip state” proposed in earlier kinetic schemes.

Revised wording:

ATP binding to any of the three symmetry-equivalent inter-subunit pockets preferentially lowers the rotational barrier of subunit A, while subunit B is only marginally affected—or even slightly disfavoured. The early rotation of subunit A distorts the neighbouring pocket it also helps form, raising the free-energy barrier for ATP binding there and thus establishing a structural basis for negative cooperativity.

Methods / Kinetic model design (new sentence to insert)

In all nine schemes, the three inter-subunit ATP pockets share a single association-rate constant (k_{on}) for the unrotated state; thus the initial collision probability is fully symmetric and any asymmetry emerges only after ligand binding.

Page 9, fourth paragraph (first sentence) Original wording:

This DCOT/DCOD analysis indicates a pronounced structural asymmetry in the receptor’s initial response to ATP binding, consistent with asymmetric coupling inferred from the kinetic analysis.

Revised wording:

DCOT/DCOD analysis shows that, once ATP is bound, the receptor’s first conformational response is markedly asymmetric—fully consistent with the post-binding asymmetric coupling deduced from the kinetic analysis.

Results / MD simulation description

Original wording:

To resolve the ambiguity in the binding–rotation coupling parameters identified by kinetic modeling, we analyzed Molecular Dynamics (MD) simulations of a closed-state P2X receptor with ATP bound at a single interface (chains A and B).

Revised wording:

To resolve the ambiguity in the binding–rotation coupling parameters identified by kinetic modeling, we ran MD simulations of the closed-state receptor with ATP pre-positioned in one arbitrarily chosen inter-subunit pocket (A–B interface). As all three pockets are energetically equivalent before binding, the interface selection carries no mechanistic bias.

Besides, we introduced the following paragraph:

The distinct LB displacements—which are structurally linked to the transmembrane domains [8]—indicate that, after ATP binds to any equivalent inter-subunit pocket, channel activation proceeds through an asymmetric conformational step initiated on the subunit located on the clockwise face of the occupied interface.

These revisions clarify that **binding symmetry is enforced** at the collision step and that all observed asymmetries derive from *post-binding* structural coupling, directly addressing the reviewer’s concerns without further simulation. We trust these changes will prevent similar misunderstandings and improve the manuscript’s clarity.

Reviewer 3 Comment 4

Comment: “4. Prior evidence indicates that at least two ATP molecules must bind for P2X activation, which supports an asymmetric coupling mechanism. Nonetheless, the authors do not adequately address the sequence and dynamics of these events— whether ATP binding occurs concurrently or sequentially. Incorporating these details into the model would enhance its explanatory power and fidelity.”

Response:

Concurrent vs. sequential ATP binding and subunit rotation

1. *Binding is intrinsically stochastic.* Two ATPs can never meet the receptor *at the exact same instant*; the average interval between the first and second binding event is $\tau_{\text{gap}} = (k_{\text{on}}[\text{ATP}])^{-1}$. Using the posterior medians $k_{\text{on}} = 6.0 \mu\text{M}^{-1}\text{s}^{-1}$ and $r_{\text{on}} = 1,680 \text{ s}^{-1}$ (rotational rate) :

- 10 mM ATP ($[\text{ATP}] = 10,000 \mu\text{M}$): $\tau_{\text{gap}} \approx 17 \mu\text{s} \ll 1/r_{\text{on}}$ (0.6 ms) \Rightarrow **practically concurrent.**

- 0.1 mM ATP ($[ATP] = 100 \mu\text{M}$): $\tau_{\text{gap}} \approx 1.7 \text{ ms} > 1/r_{\text{on}} \Rightarrow$ clearly **sequential**.

Thus the same kinetic scheme naturally reproduces both regimes without ad-hoc toggles.

2. *Model implementation.* Each binding pocket follows mass–action kinetics, so the sequence of occupancies emerges from concentration alone. During 10 mM pulses the fitted model indeed shows that ligand binding precedes subunit rotation.
3. *Concerted vs. stepwise rotation.* The nine kinetic variants differ in how they treat rotation:
 - **Schemes I–VI** implicitly or explicitly assume a channel-level, concerted rotation (flip/open) of the three subunits.
 - **Schemes VII–IX** allow each subunit to rotate independently, i.e. sequentially.

Bayesian evidence decisively favors the asymmetric, stepwise Scheme IX over its symmetric counterpart (VIII) and the all concerted schemes. However it is worth noticing that the concerted Scheme VI is close to Scheme IX in Evidence.

Minor Points:

Reviewer 3 Comment 5

Comment: “5. The authors rely on zebrafish P2X4 structures (PDB: 3I5D and 4DW1) to represent closed and open states. However, 3I5D is outdated and exhibits local inaccuracies (3.5 Å resolution), having been superseded by 4WD0 (apo, 2.9 Å). The rationale for this choice remains unclear.”

Response:

The initial crystallographic structure of the P2X receptor was reported for the closed state of the zebrafish P2X4 receptor (PDB IDs 3I5D and 3H9V). Shortly thereafter, the same group published additional structures representing both closed and open states (PDB IDs 4DW0 and 4DW1). In this later work, the authors compared the newer closed state structure (4DW0) with the earlier one (3I5D), noting an improvement in resolution from 3.1 Å to 2.9 Å. They also identified a poorly resolved region in the 3I5D map (residues 88–97), which was modeled differently in the 4DW0 structure (see Supplementary Fig. 5 in Ref. [8]). However, this difference is unlikely to significantly affect the overall receptor structure or its functional dynamics for several reasons:

- The region is located within the core of the extracellular vestibule, a relatively rigid region that is uncorrelated with subdomains motions involved in the closed-to-open transition.

- The affected segment is very small relative to the overall receptor architecture.
- The conformation change between the models (4DW0 and 3I5D) is minimal.
- Our extensive investigations of P2X receptor dynamics (using 3I5D and other crystal structures as starting models) have undergone rigorous validation, confirming their stability and robustness while revealing consistent dynamical patterns [30-33].

This extensive prior usage and validation motivated our selection of the 3I5D structure to model the closed state, considering it an appropriate and well-established template for examining the closed-state dynamics of the P2X₄ receptor in our study. The rationale for this choice is now explicitly detailed in the revised manuscript and was presented in **Comment 1**

Reviewer 3 Comment 6

Comment: “6. P2X2 specifically binds ATP^{4-} and not MgATP, with an EC50 in the range of 3–10 μ M. The use of 0.1–10 mM MgATP—well beyond saturating concentrations—is not scientifically justified.”

Response: We addressed the reviewer concerns in a new section in methods:

Electrophysiological Data and Preprocessing

We re-analyzed outside-out patch recordings from our previous study, obtained from HEK293 cells expressing rat P2X2 receptors. The dataset includes macroscopic responses to ATP pulses (0.1–10 mM, 0.2 or 10 ms duration), interleaved with 10 ms applications of 1 mM MgATP.

Active agonist and concentration rationale. All nominal concentrations refer to *total* MgATP; only the free tetravalent species (ATP^{4-}) gates the channel. At pH 7.3 with 1 mM Mg^{2+} with $K_D^{MgATP} \approx 0.2$ mM ~ 5 –8% of total MgATP is free ATP^{4-} . For the 0.2 ms pulses the system is probed in a strict pre-equilibrium regime where the probability that a binding pocket is occupied is

$$P_{\text{bind}} \approx 1 - \exp[-k_{\text{on}}[ATP^{4-}] t].$$

With the posterior median $k_{\text{on}} = 6 \mu\text{M}^{-1}\text{s}^{-1}$ (Table S3) and $t = 0.0002$ s, half-maximal occupancy already requires $[ATP^{4-}] \gtrsim 0.6$ mM—i.e. 5–10 mM total MgATP. Therefore the used 0.1–10 mM MgATP span sub-saturating to near-saturating pre-equilibrium binding.

Current traces were segmented into baseline, activation, and deactivation epochs, each resampled with exponentially increasing time steps to capture both rapid and slow kinetics. Baseline noise was quantified over 13.7 ms and 0.02 ms windows. **Full segmentation criteria, interval selection, and the preprocessing workflow are detailed in Supplementary Information Section S1.1.**

Reviewer 3 Comment 7

Comment: “7. The ATP-binding interface spans two subunits, comprising the head and LF domains of one and the DF and LB regions of the other. ATP binding induces conformational changes in both subunits, particularly in the LF, DF, and head loop regions. Yet the authors propose that only the left subunit (A) undergoes significant rotational motion, while the right subunit (B) remains largely static—a conclusion not convincingly substantiated by the data presented.”

Response:

Once both the closed and open structure of P2X were disclosed, a conformational change induced by the binding of three ATP molecules was inferred [8]. These conformations represent the initial and final states of the whole opening mechanism. What remains unknown is how the binding of one or two ATP molecules affects the structures of each of the three subunits comprising the channel. These intermediate states have not been experimentally resolved, and in such cases, molecular dynamics (MD) simulations can provide valuable insights.

To address this knowledge gap, we performed MD simulations in which a single ATP molecule was placed in one binding site of the closed form—corresponding to the first step of the kinetic mechanism. Within the simulated time window, we observed immediate local perturbations caused by the binding of a single ATP. Specifically, certain regions in subunit A exhibited rapid conformational changes, whereas subunit B, presents a more modest displacement. This asymmetric perturbation suggests that the additional conformational rearrangements required to reach the fully open state with three bound ATP molecules occur: over longer timescales (not accessible within our MD simulations) or require the sequential binding of the second and third ATP molecule.

Although the investigation of such longer-timescale events or partially occupied states lies beyond the scope of the present study, results from ten independent simulations consistently indicated that the initial perturbation at a single binding site triggered asymmetric movements of the subunits forming that site, directed toward the open conformation. These findings are in agreement with MacroIR model results, which also predicts that the binding of a single ATP induces asymmetric conformational changes in the receptor subunits.

Finally, we acknowledge that this point was not clearly articulated in the original manuscript and thank the reviewer for bringing it to our attention. In the revised version, we have refined the presentation of the results to clarify our interpretation and explicitly delineate the limitations within which our conclusions apply. All related changes and additions are next described.

- Methods:

Original wording:

We investigated the structural basis of asymmetric gating in P2X receptors using molecular dynamics (MD) simulations. Models were constructed for the zebrafish P2X4 closed and open states (PDB: 3I5D and 4DW1), with added or modeled residues to ensure full-length chains and correct symmetry. An ATP molecule was docked into the A–B binding pocket of the closed state to create a singly liganded model. Simulations were run in explicit solvent using AMBER, followed by domain-wise analysis of conformational displacements projected onto closed→open transition vectors for key extracellular regions (head loop, LF, DF, LB). Details of system preparation, simulation protocols, and vector analysis are provided in Supplementary Information, Section S1.5.

Revised wording:

To investigate the structural perturbations induced by the initial binding of a single ATP molecule to P2X, we performed molecular dynamics (MD) simulations on the closed conformation of the channel, with an agonist molecule docked into the A–B binding pocket. For this purpose, we used the crystal structure with PDB code 3I5D, which represents the first structure resolved for this channel and has been extensively analyzed by our group. In parallel, to quantify the movement of each of the three subunits toward the open conformation, we constructed a model of the open state based on PDB: 4DW1. MD simulations were carried out in explicit solvent using the AMBER package, followed by domain-wise analysis of conformational changes projected onto closed-to-open transition vectors for four key extracellular regions: the Head-loop, Left Flipper (LF), Dorsal Fin (DF), and Lower Body (LB). Further details on system preparation, simulation protocols, and vector analysis are provided in Supplementary Information, Section S1.5.

- Results:

Original wording:

To resolve the ambiguity in the binding–rotation coupling parameters identified by kinetic modeling, we analyzed Molecular Dynamics (MD) simulations of a

closed-state P2X receptor with ATP bound at a single interface (chains A and B). For each structure in the ensemble, we quantified its Degree of Closed-to-Open Transition (DCOT) and its alignment with the opening direction (Degree of Closed-to-Open Direction, DCOD).

Figure 3 presents the DCOT analysis revealing the distinct behavioral patterns between the chains A and B. Among the four regions analyzed, the results for Dorsal Fin (DF) of chain B may initially seem counterintuitive, as the DCOT values shift opposite to the expected closed→open direction. This region forms part of the ATP-occupied binding cleft in $P2X_{1-ATP}^{closed}$, and its movement is strongly correlated with the Left Flipper (LF) of chain A [18,19]. Notably, LF(A) exhibits a distribution markedly shifted toward the open conformation. This shift indicates a substantial alteration in its interaction pattern with DF(B), effectively triggering an initial "release" of DF(B) from the ATP-bound cleft and promoting movement in the direction opposite to the typical closed→open transition. In addition to the shift in LF distributions favoring chain A toward conformations closer to the open state, the Head-loop and LB regions further emphasize this asymmetry, with chain A consistently adopting more open-like conformations. Particularly, the distinct displacements observed in the LB regions of both chains, which are structurally connected to the transmembrane domains [8], suggest an initial asymmetrical channel activation mechanism triggered by the first ATP-binding event.

Revised wording:

To resolve the ambiguity in the binding–rotation coupling parameters identified by kinetic modeling, we analyzed Molecular Dynamics (MD) simulations of a closed-state P2X receptor with ATP bound at a single interface (chains A and B). These simulations aimed to infer the initial perturbations caused by one ATP molecule. For the structures sampled during the simulations, we quantified its Degree of Closed-to-Open Transition (DCOT) and its alignment with the opening direction (Degree of Closed-to-Open Direction, DCOD).

Figure 3 presents the DCOT analysis revealing the distinct behavioral patterns between the chains A and B triggered by the interaction of ATP in the binding cleft they composed. In general, it is observed that the examined regions of chain A move towards the open form in a higher degree than those of the B chain. The comparison between LF(A) and LF(B) (Fig. 3 panel b)), for instance, shows that the first one exhibits a distribution markedly shifted toward the open conformation than the latter. This shift of LF(A) generates a substantial alteration in its interaction pattern with DF(B), effectively triggering an initial "release" of DF(B) from the ATP-bound cleft and promoting a movement in the direction opposite to the typical closed→open transition (Fig. 3 panel c)). In addition to the shift in LF distributions favoring chain A toward conformations closer to the open state, the Head-loop and Lower Body (LB)

regions further emphasize this asymmetry, with chain A consistently adopting a more open-like conformations (panels a) and d) of Fig. 3. Particularly, the distinct displacements observed in the LB regions of both chains, which are structurally connected to the transmembrane domains [8], suggest an initial asymmetrical channel activation mechanism triggered by the first ATP-binding event.

Structurally, ligand-gated ion channels (LGICs) typically assemble as trimeric, tetrameric, or pentameric complexes. This study focuses on the ATP-gated P2X receptor, a prototypical trimeric channel, to explore the mechanisms of allosteric activation. While static, high-resolution structures have delineated the closed and open conformations, the cooperative interactions among the three ATP-binding events remain insufficiently understood. To address this gap, the authors employ a MacroIR-based approach in combination with molecular dynamics simulations and propose a model featuring directional and asymmetric coupling. While the findings offer valuable mechanistic insights into P2X receptor gating, the study is constrained by several notable methodological limitations.

1. The authors utilize molecular dynamics simulations based on P2X receptor structures; however, they chose the structurally unresolved P2X2 subtype for electrophysiological experiments, despite the availability of high-resolution structures for P2X1, P2X3, P2X4, and P2X7. Similarly, for ATP-binding simulations, they employ the zebrafish P2X4 (zfP2X4) structure, rather than P2X3 or P2X7, which offer both open and closed conformations. These choices lack sufficient scientific justification.
2. LGICs span a range of oligomeric forms—trimeric (e.g., P2X, ASIC), tetrameric (e.g., TRP, Kv, NMDA), and pentameric (e.g., nAChR, 5-HTR)—with varying degrees of structural symmetry. A twist-to-open gating mechanism involving cooperative transitions has been proposed for several subtypes. The current study introduces an asymmetric, negatively cooperative activation model for P2X2, challenging the canonical paradigm of symmetric, cooperative gating. However, extrapolating these findings to other LGICs is premature and potentially misleading, as trimeric receptors cannot be assumed to reflect the mechanistic diversity of all LGIC families.
3. Multiple ATP-binding scenarios are plausible for P2X2. The authors propose a model in which one site exhibits a higher probability of ATP collision, initiating allosteric modulation of the remaining sites. However, they fail to consider an alternative, equally plausible model involving equivalent binding probabilities and cooperative dual-site engagement. Their reliance on a single ATP-binding simulation using zfP2X4 (apo-ATP) lacks sufficient resolution to exclude such alternatives.
4. Prior evidence indicates that at least two ATP molecules must bind for P2X activation, which supports an asymmetric coupling mechanism. Nonetheless, the authors do not adequately address the sequence and dynamics of these events—whether ATP binding occurs concurrently or sequentially. Incorporating these details into the model would enhance its explanatory power and fidelity.

Minor Points:

5. The authors rely on zebrafish P2X4 structures (PDB: 3I5D and 4DW1) to represent closed and open states. However, 3I5D is outdated and exhibits local inaccuracies (3.5 Å resolution), having been superseded by 4WD0 (apo, 2.9 Å). The rationale for this choice remains unclear.

6. P2X2 specifically binds ATP⁴⁻ and not MgATP, with an EC₅₀ in the range of 3–10 μM. The use of 0.1–10 mM MgATP—well beyond saturating concentrations—is not scientifically justified.
7. The ATP-binding interface spans two subunits, comprising the head and LF domains of one and the DF and LBD regions of the other. ATP binding induces conformational changes in both subunits, particularly in the LF, DF, and head loop regions. Yet the authors propose that only the left subunit (A) undergoes significant rotational motion, while the right subunit (B) remains largely static—a conclusion not convincingly substantiated by the data presented.